# Creating and controlling visual environments using BonVision

**Gonçalo Lopes[1], Karolina Farrell[2‡], Edward AB Horrocks[2‡], Chi-Yu Lee[2‡], Mai M Morimoto[2‡], Tomaso Muzzu[2‡], Amalia Papanikolaou[2‡], Fabio R Rodrigues[2‡], Thomas Wheatcroft[2‡], Stefano Zucca[2‡], Samuel G Solomon[2†*], Aman B Saleem[2†*]**

[1]NeuroGEARS Ltd., London, United Kingdom; [2]UCL Institute of Behavioural Neuroscience, Department of Experimental Psychology, University College London, London, United Kingdom

**Abstract** Real-time rendering of closed-loop visual environments is important for next-generation understanding of brain function and behaviour, but is often prohibitively difficult for non-experts to implement and is limited to few laboratories worldwide. We developed BonVision as an easy-to-use open-source software for the display of virtual or augmented reality, as well as standard visual stimuli. BonVision has been tested on humans and mice, and is capable of supporting new experimental designs in other animal models of vision. As the architecture is based on the open-source Bonsai graphical programming language, BonVision benefits from native integration with experimental hardware. BonVision therefore enables easy implementation of closed-loop experiments, including real-time interaction with deep neural networks, and communication with behavioural and physiological measurement and manipulation devices.

**\*For correspondence:**
s.solomon@ucl.ac.uk (SGS);
aman.saleem@ucl.ac.uk (ABS)

†These authors contributed
equally to this work
‡These authors are listed
alphabetically

**Competing interest:** See
page 11

**Reviewing editor:** Chris I Baker,
National Institute of Mental
Health, National Institutes of
Health, United States

## Introduction

Understanding behaviour and its underlying neural mechanisms calls for the ability to construct and control environments that immerse animals, including humans, in complex naturalistic environments that are responsive to their actions. Gaming-driven advances in computation and graphical rendering have driven the development of immersive closed-loop visual environments, but these new platforms are not readily amenable to traditional research paradigms. For example, they do not specify an image in egocentric units (of visual angle), sacrifice precise control of a visual display, and lack transparent interaction with external hardware.

Most vision research has been performed in non-immersive environments with standard two-dimensional visual stimuli, such as gratings or dot stimuli, using established platforms including PsychToolbox (*Brainard, 1997*) or PsychoPy (*Peirce, 2007*; *Peirce, 2008*). Pioneering efforts to bring gaming-driven advances to neuroscience research have provided new platforms for closed-loop visual stimulus generation: STYTRA (*Štih et al., 2019*) provides 2D visual stimuli for larval zebrafish in python, ratCAVE (*Del Grosso and Sirota, 2019*) is a specialised augmented reality system for rodents in python, FreemoVR (*Stowers et al., 2017*) provides virtual reality in Ubuntu/Linux, and ViRMEn (*Aronov and Tank, 2014*) provides virtual reality in Matlab. However, these new platforms lack the generalised frameworks needed to specify or present standard visual stimuli.

Our initial motivation was to create a visual display software with three key features. First, an integrated, standardised platform that could rapidly switch between traditional visual stimuli (such as grating patterns) and immersive virtual reality. Second, the ability to replicate experimental workflows across different physical configurations (e.g. when moving from one to two computer monitors, or from flat-screen to spherical projection). Third, the ability for rapid and efficient interfacing with external hardware (needed for experimentation) without needing to develop complex multi-

threaded routines. We wanted to provide these advances in a way that made it easier for users to construct and run closed-loop experimental designs. In closed-loop experiments, stimuli are ideally conditioned by asynchronous inputs, such as those provided by multiple independent behavioural and neurophysiological measurement devices. Most existing platforms require the development of multi-threaded routines to run experimental paradigms (e.g. control brain stimulation, or sample from recording devices) without compromising the rendering of visual scenes. Implementing such multi-thread routines is complex. We therefore chose to develop a visual presentation framework within the Bonsai programming language (*Lopes et al., 2015*). Bonsai is a graphical, high-performance, and event-based language that is widely used in neuroscience experiments and is already capable of real-time interfacing with most types of external hardware. Bonsai is specifically designed for flexible and high-performance composition of data streams and external events, and is therefore able to monitor and connect multiple sensor and effector systems in parallel, making it easier to implement closed-loop experimental designs.

We developed BonVision, an open-source software package that can generate and display well-defined visual stimuli in 2D and 3D environments. BonVision exploits Bonsai's ability to run OpenGL commands on the graphics card through the Bonsai.Shaders package. BonVision further extends Bonsai by providing pre-built GPU shaders and resources for stimuli used in vision research, including movies, along with an accessible, modular interface for composing stimuli and designing experiments. The definition of stimuli in BonVision is independent of the display hardware, allowing for easy replication of workflows across different experimental configurations. Additional unique features include the ability to automatically detect and define the relationship between the observer and the display from a photograph of the experimental apparatus, and to use the outputs of real-time inference methods to determine the position and pose of an observer online, thereby generating augmented reality environments.

## Results

To provide a framework that allowed both traditional visual presentation and immersive virtual reality, we needed to bring these very different ways of defining the visual scene into the same architecture. We achieved this by mapping the 2D retino-centric coordinate frame (i.e. degrees of the visual field) to the surface of a 3D sphere using the Mercator projection (*Figure 1A*, *Figure 1—figure supplement 1*). The resulting sphere could therefore be rendered onto displays in the same way as any other 3D environment. We then used 'cube mapping' to specify the 360° projection of 3D environments onto arbitrary viewpoints around an experimental observer (human or animal; *Figure 1B*). Using this process, a display device becomes a window into the virtual environment, where each pixel on the display specifies a vector from the observer through that window. The vector links pixels on the display to pixels in the 'cube map', thereby rendering the corresponding portion of the visual field onto the display.

Our approach has the advantage that the visual stimulus is defined irrespectively of display hardware, allowing us to independently define each experimental apparatus without changing the preceding specification of the visual scene, or the experimental design (*Figure 1C–E*, *Figure 1—figure supplements 1* and *2*). Consequently, BonVision makes it easy to replicate visual environments and experimental designs on various display devices, including multiple monitors, curved projection surfaces, and head-mounted displays (*Figure 1C–E*). To facilitate easy and rapid porting between different experimental apparatus, BonVision features a fast semi-automated display calibration. A photograph of the experimental setup with fiducial markers (*Garrido-Jurado et al., 2014*) measures the 3D position and orientation of each display relative to the observer (*Figure 2* and *Figure 2—figure supplement 1*). BonVision's inbuilt image processing algorithms then estimate the position and orientation of each marker to fully specify the display environment.

Virtual reality environments are easy to generate in BonVision. BonVision has a library of standard pre-defined 3D structures (including planes, spheres, and cubes), and environments can be defined by specifying the position and scale of the structures, and the textures rendered on them (e.g. *Figure 1—figure supplement 2* and Figure 5F). BonVision also has the ability to import standard format 3D design files created elsewhere in order to generate more complex environments (file formats listed in Materials and methods). This allows users to leverage existing 3D drawing platforms

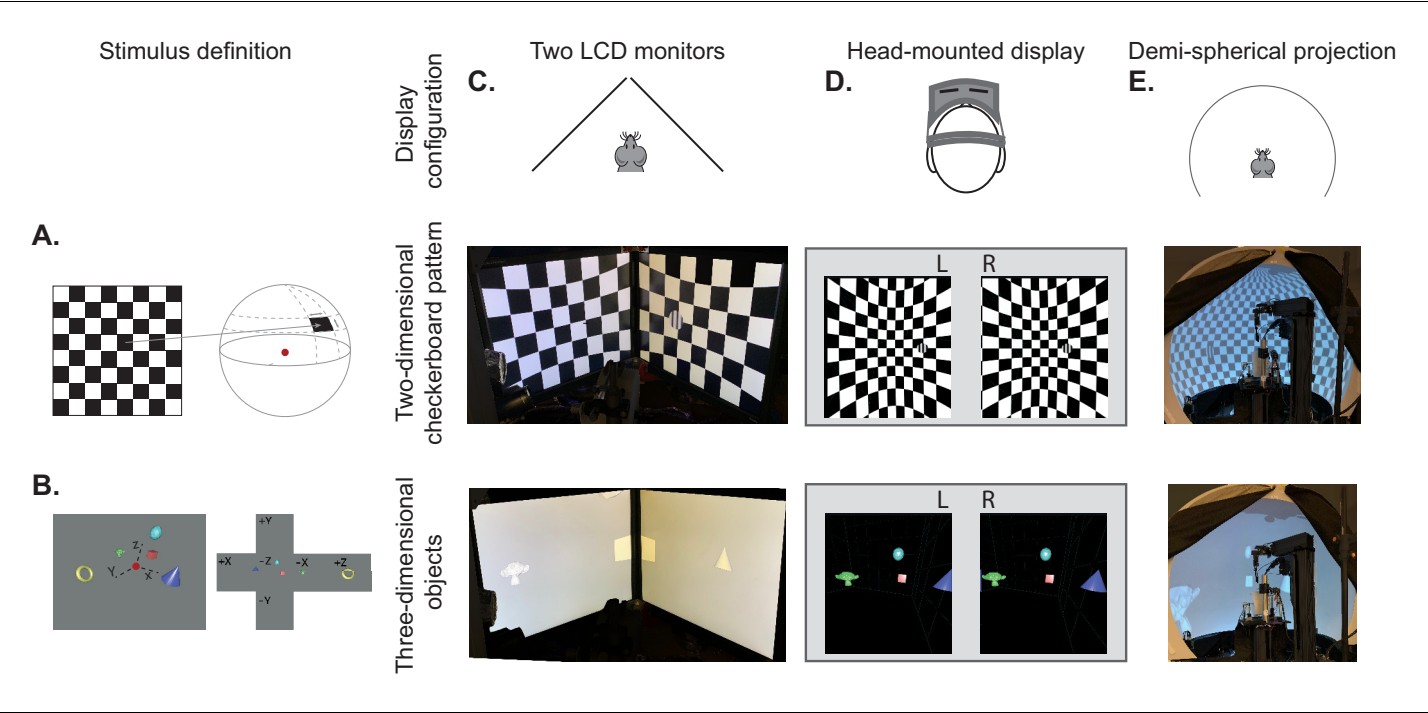

**Figure 1.** BonVision's adaptable display and render configurations. (**A**) Illustration of how two-dimensional textures are generated in BonVision using Mercator projection for sphere mapping, with elevation as latitude and azimuth as longitude. The red dot indicates the position of the observer. (**B**) Three-dimensional objects were placed at the appropriate positions and the visual environment was rendered using cube-mapping. (**C–E**) Examples of the same two stimuli, a checkerboard + grating (middle row) or four three-dimensional objects (bottom row), displayed in different experimental configurations (top row): two angled LCD monitors (**C**), a head-mounted display (**D**), and demi-spherical dome (**E**).
The online version of this article includes the following figure supplement(s) for figure 1:

**Figure supplement 1.** Mapping stimuli onto displays in various positions.

**Figure supplement 2.** Modular structure of workflow and example workflows.

(including open source platform 'Blender': https://www.blender.org/) to construct complex virtual scenes (see Appendix 1).

BonVision can define the relationship between the display and the observer in real-time. This makes it easy to generate augmented reality environments, where what is rendered on a display depends on the position of an observer (*Figure 3A*). For example, when a mouse navigates through an arena surrounded by displays, BonVision enables closed-loop, position-dependent updating of those displays. Bonsai can track markers to determine the position of the observer, but it also has turn-key capacity for real-time live pose estimation techniques – using deep neural networks (*Mathis et al., 2018*; *Pereira et al., 2019*; *Kane et al., 2020*) – to keep track of the observer's movements. This allows users to generate and present interactive visual environments (simulation in *Figure 3—video 1* and *Figure 3B and C*).

BonVision is capable of rendering visual environments near the limits of the hardware (*Figure 4*). This is possible because Bonsai is based on a just-in-time compiler architecture such that there is little computational overhead. BonVision accumulates a list of the commands to OpenGL as the programme makes them. To optimise rendering performance, the priority of these commands is ordered according to that defined in the Shaders component of the *LoadResources* node (which the user can manipulate for high-performance environments). These ordered calls are then executed when the frame is rendered. To benchmark the responsiveness of BonVision in closed-loop experiments, we measured the delay (latency) between an external event and the presentation of a visual stimulus. We first measured the closed-loop latency for BonVision when a monitor was refreshed at a rate of 60 Hz (*Figure 4A*). We found delays averaged 2.11 ± 0.78 frames (35.26 ± 13.07 ms). This latency was slightly shorter than that achieved by PsychToolbox (*Brainard, 1997*) on the same laptop (2.44 ± 0.59 frames, 40.73 ± 9.8 ms; Welch's t-test, $p<10^{-80}$, n = 1000). The overall latency of

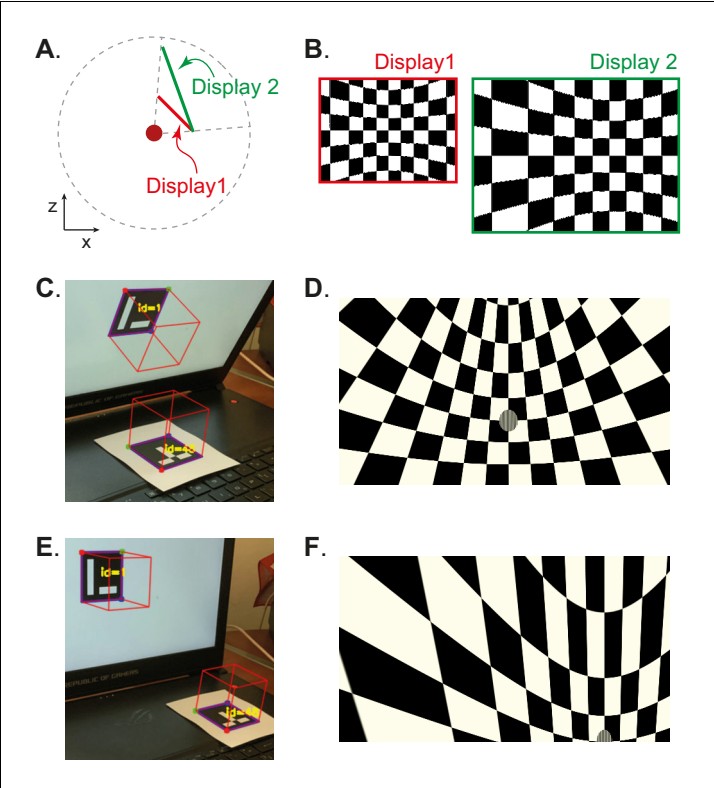

**Figure 2.** Automated calibration of display position. (**A**) Schematic showing the position of two hypothetical displays of different sizes, at different distances and orientation relative to the observer (red dot). (**B**) How a checkerboard of the same visual angle would appear on each of the two displays. (**C**) Example of automatic calibration of display position. Standard markers are presented on the display, or in the environment, to allow automated detection of the position and orientation of both the display and the observer. These positions and orientations are indicated by the superimposed red cubes as calculated by BonVision. (**D**) How the checkerboard would appear on the display when rendered, taking into account the precise position of the display.

(**E and F**) Same as (**C and D**), but for another pair of display and observer positions. The automated calibration was based on the images shown in **C** and **E**.

The online version of this article includes the following figure supplement(s) for figure 2:

**Figure supplement 1.** Automated workflow to calibrate display position.

**Figure supplement 2.** Automated gamma-calibration of visual displays.

BonVision was mainly constrained by the refresh rate of the display device, such that higher frame rate displays yielded lower latency (60 Hz: 35.26 ± 13.07 ms; 90 Hz: 28.45 ± 7.22 ms; 144 Hz: 18.49 ± 10.1 ms; *Figure 4A*). That is, the number of frames between the external event and stimulus presentation was similar across frame rate (60 Hz: 2.11 ± 0.78 frames; 90 Hz: 2.56 ± 0.65 frames; 144 Hz: 2.66 ± 1.45 frames; *Figure 4C*). We used two additional methods to benchmark visual display performance relative to other frameworks (we did not try to optimise code fragments for each framework) (*Figure 4B and C*). BonVision was able to render up to 576 independent elements and up to eight overlapping textures at 60 Hz without missing ('dropping') frames, broadly matching Psy-choPy (*Peirce, 2007*; *Peirce, 2008*) and Psychtoolbox (*Brainard, 1997*). BonVision's performance was similar at different frame rates – at standard frame rate (60 Hz) and at 144 Hz (*Figure 4—figure supplement 1*). BonVision achieved slightly fewer overlapping textures than PsychoPy, as BonVision does not currently have the option to trade-off the resolution of a texture and its mask for performance. BonVision also supports video playback, either by preloading the video or by streaming it from the disk. The streaming mode, which utilises real-time file I/O and decompression, is capable of displaying both standard definition (SD: 480 p) and full HD (HD: 1080 p) at 60 Hz on a standard computer (*Figure 4D*). At higher rates, performance is impaired for Full HD videos, but is improved by buffering, and fully restored by preloading the video onto memory (*Figure 4D*). We benchmarked

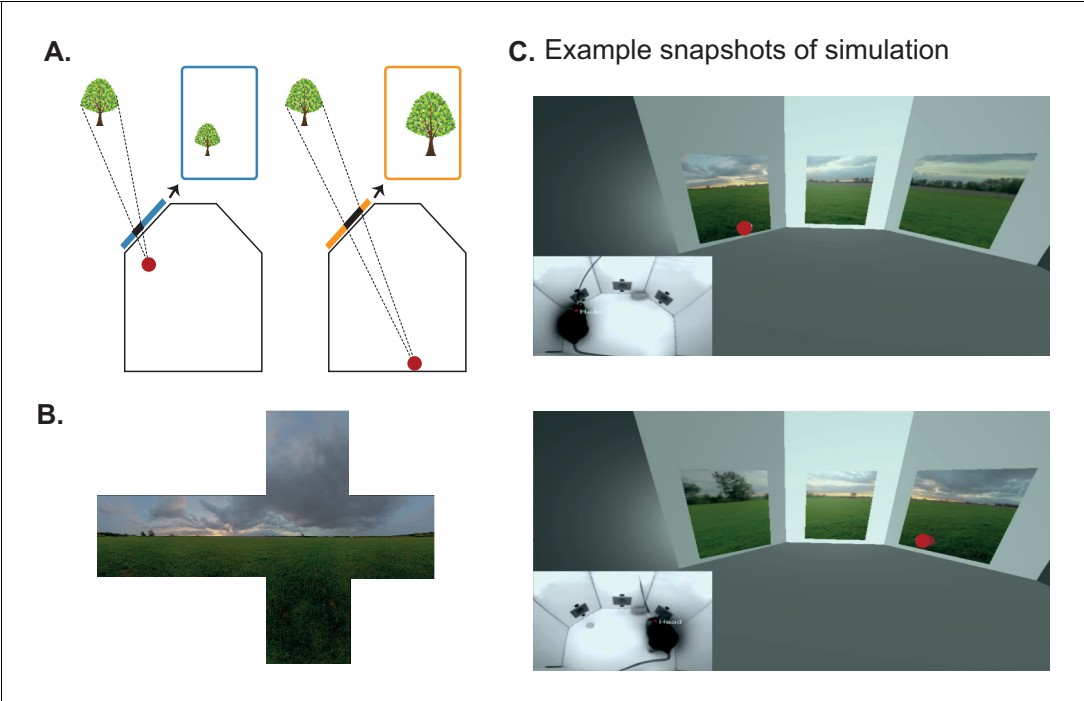

**Figure 3.** Using BonVision to generate an augmented reality environment. (**A**) Illustration of how the image on a fixed display needs to adapt as an observer (red dot) moves around an environment. The displays simulate windows from a box into a virtual world outside. (**B**) The virtual scene (from: http://scmapdb.com/wad:skybox-skies) that was used to generate the example images and *Figure 3—video 1* offline. (**C**) Real-time simulation of scene rendering in augmented reality. We show two snapshots of the simulated scene rendering, which is also shown in *Figure 3—video 1*. In each case the inset image shows the actual video images, of a mouse exploring an arena, that were used to determine the viewpoint of an observer in the simulation. The mouse's head position was inferred (at a rate of 40 frames/s) by a network trained using DeepLabCut (*Aronov and Tank, 2014*). The top image shows an instance when the animal was on the left of the arena (head position indicated by the red dot in the main panel) and the lower image shows an instance when it was on the right of the arena.

The online version of this article includes the following video for figure 3:

**Figure 3—video 1.** Augmented reality simulation using BonVision.

https://elifesciences.org/articles/65541#fig3video1

BonVision on a standard Windows OS laptop, but BonVision is now also capable of running on Linux.

To confirm that the rendering speed and timing accuracy of BonVision are sufficient to support neurophysiological experiments, which need high timing precision, we mapped the receptive fields of neurons early in the visual pathway (*Yeh et al., 2009*), in the mouse primary visual cortex and superior colliculus. The stimulus ('sparse noise') consisted of small black or white squares briefly (0.1 s) presented at random locations (*Figure 5A*). This stimulus, which is commonly used to measure receptive fields of visual neurons, is sensitive to the timing accuracy of the visual stimulus, meaning that errors in timing would prevent the identification of receptive fields. In our experiments using BonVision, we were able to recover receptive fields from electrophysiological measurements - both in the superior colliculus and primary visual cortex of awake mice (*Figure 5B and C*) – demonstrating that BonVision meets the timing requirements for visual neurophysiology. The receptive fields show in *Figure 5C* were generated using timing signals obtained directly from the stimulus display (via a photodiode). BonVision's independent logging of stimulus presentation timing was also sufficient to capture the receptive field (*Figure 5—figure supplement 1*).

To assess the ability of BonVision to control virtual reality environments we first tested its ability to present stimuli to human observers on a head-mounted display (*Scarfe and Glennerster, 2015*). BonVision uses positional information (obtained from the head-mounted display) to update the view of the world that needs to be provided to each eye, and returns two appropriately rendered images. On each trial, we asked observers to identify the larger of two non-overlapping cubes that were

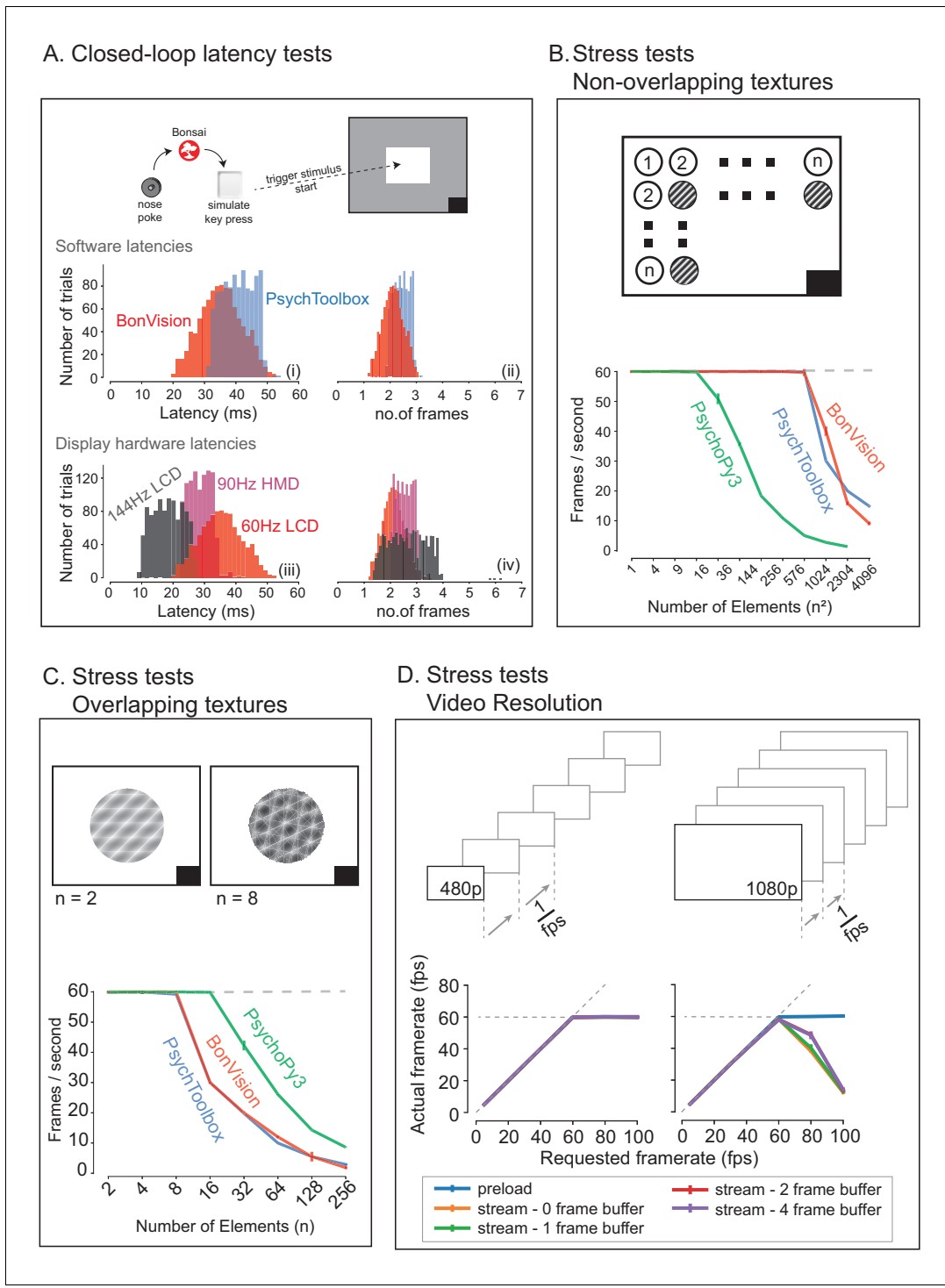

**Figure 4.** Closed-loop latency and performance benchmarks. (**A**) Latency between sending a command (virtual key press) and updating the display (measured using a photodiode). (**A.i and A.ii**) Latency depended on the frame rate of the display, updating stimuli with a delay of one to three frames. (**A.iii and A.iv**). (**B and C**) Benchmarked performance of BonVision with respect to Psychtoolbox and PsychoPy. (**B**) When using non-overlapping textures BonVision and Psychtoolbox could present 576 independent textures without dropping frames, while PsychoPy could present 16. (**C**) When using overlapping textures PsychoPy could present 16 textures, while BonVision and Psychtoolbox could present eight textures without dropping frames. (**D**) Benchmarks for movie playback. BonVision is capable of displaying standard definition (480 p) and high definition (1080 p) movies at 60 frames/s on a laptop computer with a standard CPU and graphics card. We measured display rate when fully pre-loading the

*Figure 4 continued on next page*

*Figure 4 continued*

movie into memory (blue), or when streaming from disk (with no buffer: orange; 1-frame buffer: green; 2-frame buffer: red; 4-frame buffer: purple). When asked to display at rates higher than the monitor refresh rate (>60 frames/s), the 480 p video played at the maximum frame rate of 60fps in all conditions, while the 1080 p video reached the maximum rate when pre-loaded. Using a buffer slightly improved performance. A black square at the bottom right of the screen in **A–C** is the position of a flickering rectangle, which switches between black and white at every screen refresh. The luminance in this square is detected by a photodiode and used to measure the actual frame flip times.

The online version of this article includes the following figure supplement(s) for figure 4:

**Figure supplement 1.** BonVision performance benchmarks at high frame rate.

placed at different virtual depths (*Figure 5D and E*). The display was updated in closed-loop to allow observers to alter their viewpoint by moving their head. Distinguishing objects of the same retinal size required observers to use depth-dependent cues (*Rolland et al., 1995*), and we found that all observers were able to identify which cube was larger (*Figure 5E*).

We next asked if BonVision was capable of supporting other visual display environments that are increasingly common in the study of animal behaviour. We first projected a simple environment onto a dome that surrounded a head-fixed mouse (as shown in *Figure 1E*). The mouse was free to run on a treadmill, and the treadmill's movements were used to update the mouse's position on a virtual platform (*Figure 5F*). Not only did mouse locomotion speed increase with repeated exposure, but

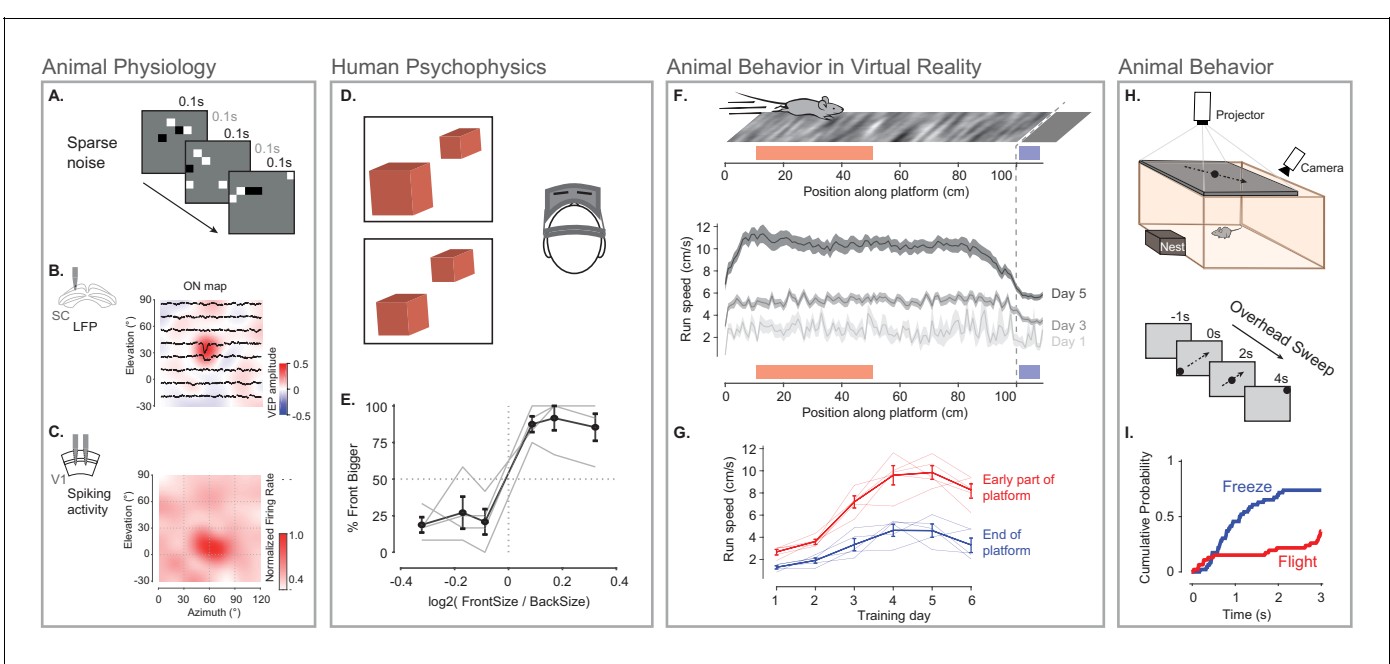

**Figure 5.** Illustration of BonVision across a range of vision research experiments. (**A**) Sparse noise stimulus, generated with BonVision, is rendered onto a demi-spherical screen. (**B and C**) Receptive field maps from recordings of local field potential in the superior colliculus (**B**), and spiking activity in the primary visual cortex (**C**) of mouse. (**D**) Two cubes were presented at different depths in a virtual environment through a head-mounted display to human subjects. Subjects had to report which cube was larger: left or right. (**E**) Subjects predominantly reported the larger object correctly, with a slight bias to report that the object in front was bigger. (**F**) BonVision was used to generate a closed-loop virtual platform that a mouse could explore (top: schematic of platform). Mice naturally tended to run faster along the platform, and in later sessions developed a speed profile, where they slowed down as they approached the end of the platform (virtual cliff). (**G**) The speed of the animal at the start of the platform and at the end of the platform as a function training. (**H**) BonVision was used to present visual stimuli overhead while an animal was free to explore an environment (which included a refuge). The stimulus was a small dot (5° diameter) moving across the projected surface over several seconds. (**I**) The cumulative probability of Freeze and Flight behaviour across time in response to moving dot presented overhead.

The online version of this article includes the following figure supplement(s) for figure 5:

**Figure supplement 1.** BonVision timing logs are sufficient to support receptive field mapping of spiking activity.

the animals modulated their speed depending on their location in the platform (*Figure 5F and G*). BonVision is therefore capable of generating virtual reality environments which both elicit and are responsive to animal behaviour. BonVision was also able to produce instinctive avoidance behaviours in freely moving mice (*Figure 5H and I*). We displayed a small black dot slowly sweeping across the overhead visual field. Visual stimuli presented in BonVision primarily elicited a freezing response, which similar experiments have previously described (*De Franceschi et al., 2016*; *Figure 5I*). Together these results show that BonVision provides sufficient rendering performance to support human and animal visual behaviour.

## Discussion

BonVision is a single software package to support experimental designs that require visual display, including virtual and augmented reality environments. BonVision is easy and fast to implement, cross-platform and open source, providing versatility and reproducibility.

BonVision makes it easier to address several barriers to reproducibility in visual experiments. First, BonVision is able to replicate and deliver visual stimuli on very different experimental apparatus. This is possible because BonVision's architecture separates specification of the display and the visual environment. Second, BonVision includes a library of workflows and operators to standardise and ease the construction of new stimuli and virtual environments. For example, it has established protocols for defining display positions (*Figure 3*), mesh-mapping of curved displays (*Figure 1E*), and automatic linearisation of display luminance (*Figure 4*), as well as a library of examples for experiments commonly used in visual neuroscience. In addition, the modular structure of BonVision enables the development and exchange of custom nodes for generating new visual stimuli or functionality without the need to construct the complete experimental paradigm. Third, BonVision is based on Bonsai (*Lopes et al., 2015*), which has a large user base and an active developer community, and is now a standard tool for open-source neuroscience research. BonVision naturally integrates Bonsai's established packages in the multiple domains important for modern neuroscience, which are widely used in applications including real-time video processing (*Zacarias et al., 2018*; *Buccino et al., 2018*), optogenetics (*Zacarias et al., 2018*; *Buccino et al., 2018*; *Moreira et al., 2019*), fibre photometry (*Soares et al., 2016*; *Hrvatin et al., 2020*), electrophysiology (including specific packages for Open Ephys *Siegle et al., 2017*; *Neto et al., 2016* and high-density silicon probes *Jun et al., 2017*; *Dimitriadis, 2018*), and calcium imaging (e.g. UCLA miniscope *Aharoni et al., 2019*; *Cai et al., 2016*). Bonsai requires researchers to get accustomed to its graphical interface and event-based framework. However, it subsequently reduces the time required to learn real-time programming, and the time to build new interfaces with external devices (see Appendix 1). Moreover, since Bonsai workflows can be called via the command line, BonVision can also be integrated into pre-existing, specialised frameworks in established laboratories.

In summary, BonVision can generate complex 3D environments and retinotopically defined 2D visual stimuli within the same framework. Existing platforms used for vision research, including PsychToolbox (*Brainard, 1997*), PsychoPy (*Peirce, 2007*; *Peirce, 2008*), STYTRA (*Štih et al., 2019*), or RigBox (*Bhagat et al., 2020*), focus on well-defined 2D stimuli. Similarly, gaming-driven software, including FreemoVR (*Stowers et al., 2017*), ratCAVE (*Del Grosso and Sirota, 2019*), and ViRMEn (*Aronov and Tank, 2014*), are oriented towards generating virtual reality environments. BonVision combines the advantages of both these approaches in a single framework (Appendix 1), while bringing the unique capacity to automatically calibrate the display environment, and use deep neural networks to provide real-time control of virtual environments. Experiments in BonVision can be rapidly prototyped and easily replicated across different display configurations. Being free, open-source, and portable, BonVision is a state-of-the-art tool for visual display that is accessible to the wider community.

## Materials and methods

### Benchmarking

We performed benchmarking to measure latencies and skipped ('dropped') frames. For benchmarks at 60 Hz refresh rate, we used a standard laptop with the following configuration: Dell Latitude

7480, Intel Core i7-6600U Processor Base with Integrated HD Graphics 520 (Dual Core, 2.6 GHz), 16 GB RAM. For higher refresh rates we used a gaming laptop ASUS ROG Zephyrus GX501GI, with an Intel Core i7-8750H (six cores, 2.20 GHz), 16 GB RAM, equipped with a NVIDIA GeForce GTX 1080. The gaming laptop's built-in display refreshes at 144 Hz, and for measuring latencies at 90 Hz we connected it to a Vive Pro SteamVR head-mounted display (90 Hz refresh rate). All tests were run on Windows 10 Pro 64-bit.

To measure the time from input detection to display update, as well as dropped frames detection, we used open-source HARP devices from Champalimaud Research Scientific Hardware Platform, using the Bonsai.HARP package. Specifically we used the HARP Behavior device (a lost latency DAQ; https://www.cf-hw.org/harp/behavior), to synchronise all measurements with the extensions: 'Photodiode v2.1' to measure the change of the stimulus on the screen, and 'Mice poke simple v1.2' as the nose poke device to externally trigger changes. To filter out the infrared noise generated from an internal LED sensor inside the Vive Pro HMD, we positioned an infrared cut-off filter between the internal headset optics and the photodiode. Typically, the minimal latency for any update is two frames: one which is needed for the VSync, and one is the delay introduced by the OS. Display hardware can add further delays if they include additional buffering. Benchmarks for video playback were carried out using a trailer from the Durian Open Movie Project ( copyright Blender Foundation | durian.blender.org).

All benchmark programmes and data are available at https://github.com/bonvision/benchmarks.

## File formats

We tested the display of images and videos using the image and video benchmark workflows. We confirmed the ability to use the following image formats: PNG, JPG, BMP, TIFF, and GIF. Movie display relies on the FFmpeg library (https://ffmpeg.org/), an industry standard, and we confirmed ability to use the following containers: AVI, MP4, OGG, OGV, and WMV; in conjunction with standard codecs: H264, MPEG4, MPEG2, DIVX. Importing 3D models and complex scenes relies on the Open Asset Importer Library (Assimp | http://assimp.org/). We confirmed the ability to import and render 3D models and scenes from the following formats: OBJ, Blender.

## Animal experiments

All experiments were performed in accordance with the Animals (Scientific Procedures) Act 1986 (United Kingdom) and Home Office (United Kingdom) approved project and personal licenses. The experiments were approved by the University College London Animal Welfare Ethical Review Board under Project License 70/8637. The mice (C57BL6 wild-type) were group-housed with a maximum of five to a cage, under a 12 hr light/dark cycle. All behavioural and electrophysiological recordings were carried out during the dark phase of the cycle.

## Innate defensive behaviour

Mice (five male, C57BL6, 8 weeks old) were placed in a 40 cm square arena. A dark refuge placed outside the arena could be accessed through a 10 cm door in one wall. A DLP projector (Optoma GT760) illuminated a screen 35 cm above the arena with a grey background (80 candela/m$^2$). When the mouse was near the centre of the arena, a 2.5 cm black dot appeared on one side of the projection screen and translated smoothly to the opposite side over 3.3 s. Ten trials were conducted over 5 days and the animal was allowed to explore the environment for 5–10 min before the onset of each trial.

Mouse movements were recorded with a near infrared camera (Blackfly S, BFS-U3-13Y3M-C, sampling rate: 60 Hz) positioned over the arena. An infrared LED was used to align video and stimulus. Freezing was defined as a drop in the animal speed below 2 cm/s that lasted more than 0.1 s; flight responses as an increase in the animal running speed above 40 cm/s (*De Franceschi et al., 2016*). Responses were only considered if they occurred within 3.5 s from stimulus onset.

## Surgery

Mice were implanted with a custom-built stainless-steel metal plate on the skull under isoflurane anaesthesia. A ~1 mm craniotomy was performed either over the primary visual cortex (2 mm lateral

and 0.5 mm anterior from lambda) or superior colliculus (0.5 mm lateral and 0.2 mm anterior from lambda). Mice were allowed to recover for 4–24 hr before the first recording session.

We used a virtual reality apparatus similar to those used in previous studies (*Schmidt-Hieber and Häusser, 2013*; *Muzzu et al., 2018*). Briefly, mice were head-fixed above a polystyrene wheel with a radius of 10 cm. Mice were positioned in the geometric centre of a truncated spherical screen onto which we projected the visual stimulus. The visual stimulus was centred at +60° azimuth and +30° elevation and had a span of 120° azimuth and 120° elevation.

## Virtual reality behaviour

Five male, 8-week-old, C57BL6 mice were used for this experiment. One week after the surgery, mice were placed on a treadmill and habituated to the virtual reality (VR) environment by progressively increasing the number of time spent head fixed: from ~15 min to 2 hr. Mice spontaneously ran on the treadmill, moving through the VR in absence of reward. The VR environment was a 100 cm long platform with a patterned texture that animals ran over for multiple trials. Each trial started with an animal at the start of the platform and ended when it reached the end, or if 60 s had elapsed. At the end of a trial, there was a 2 s grey interval before the start of the next trial.

## Neural recordings

To record neural activity, we used multi-electrode array probes with two shanks and 32 channels (ASSY-37 E-1, Cambridge Neurotech Ltd., Cambridge, UK). Electrophysiology data was acquired with an Open Ephys acquisition board connected to a different computer from that used to generate the visual stimulus.

The electrophysiological data from each session was processed using Kilosort 1 or Kilosort 2 (*Pachitariu et al., 2016*). We synchronised spike times with behavioural data by aligning the signal of a photodiode that detected the visual stimuli transitions (PDA25K2, Thorlabs, Inc, USA). We sampled the firing rate at 60 Hz, and then smoothed it with a 300 ms Gaussian filter. We calculated receptive fields as the average firing rate or local field potential elicited by the appearance of a stimulus in each location (custom routines in MATLAB).

## Augmented reality for mice

The mouse behaviour videos were acquired by Bruno Cruz from the lab of Joe Paton at the Champalimaud Centre for the Unknown, using methods similar to *Soares et al., 2016*. A *ResNet-50* network was trained using DeepLabCut (*Mathis et al., 2018*; *Kane et al., 2020*). We simulated a visual environment in which a virtual scene was presented beyond the arena, and updated the scenes on three walls of the arena. This simulated how the view changed as the animal moved through the environment. The position of the animal was updated from the video file at a rate of 40 frames/s on a gaming laptop: ASUS ROG Zephyrus GX501GI, with an Intel Core i7-8750H (six cores, 2.20 GHz), 16 GB RAM, equipped with a NVIDIA GeForce GTX 1080, using a 512 × 512 video. The performance can be improved using a lower pixel resolution for video capture, and we were able to achieve up to 80 frames/s without a noticeable decrease in tracking accuracy using this strategy. Further enhancements can be achieved using a MobileNetV2 network (*Kane et al., 2020*). The position inference from the deep neural network and the BonVision visual stimulus rendering were run on the same machine.

## Human psychophysics

All procedures were approved by the Experimental Psychology Ethics Committee at University College London (Ethics Application EP/2019/002). We obtained informed consent and consent to publish from all participants. Four male participants were tested for this experiment. The experiments were run on a gaming laptop (described above) connected to a Vive Pro SteamVR head-mounted display (90 Hz refresh rate). BonVision is compatible with different headsets (e.g. Oculus Rift, HTC Vive). BonVision receives the projection matrix (perspective projection of world display) and the view matrix (position of eye in the world) for each eye from the head set. BonVision uses these matrices to generate two textures, one for the left eye and one for the right eye. Standard onboard computations on the headset provide additional non-linear transformations that account for the relationship between the eye and the display (such as lens distortion effects).

## Code availability

BonVision is an open-source software package available to use under the MIT license. It can be downloaded through the Bonsai (bonsai-rx.org) package manager, and the source code is available at: github.com/bonvision/BonVision. All benchmark programmes and data are available at https://github.com/bonvision/benchmarks (copy archived at swh:1:rev:7205c04aa8fcba1075e9-c9991ac117bd25e92639, *Lopes, 2021*). Installation instructions, demos, and learning tools are available at: bonvision.github.io/.

## Acknowledgements

We are profoundly thankful to Bruno Cruz and Joe Paton for sharing their videos of mouse behaviour. This work was supported by a Wellcome Enrichment award: Open Research (200501/Z/16/A), Sir Henry Dale Fellowship from the Wellcome Trust and Royal Society (200501), Human Science Frontiers Program grant (RGY0076/2018) to ABS, an International Collaboration Award (with Adam Kohn) from the Stavros Niarchos Foundation/Research to Prevent Blindness to SGS, Medical Research Council grant (R023808), Biotechnology and Biological Sciences Research Council grant (R004765) to SGS and ABS.

## Additional information

### Competing interests

Gonçalo Lopes: Gonçalo Lopes is affiliated with NeuroGEARS Ltd. The author has no financial interests to declare. The other authors declare that no competing interests exist.

### Funding

| Funder | Grant reference number | Author |
|---|---|---|
| Wellcome Trust | 200501/Z/16/A | Aman B Saleem |
| Wellcome Trust | Sir Henry Dale Fellowship (200501) | Aman B Saleem |
| Royal Society | Sir Henry Dale Fellowship (200501) | Aman B Saleem |
| Medical Research Council | R023808 | Samuel G Solomon Aman B Saleem |
| Stavros Niarchos Foundation | | Samuel G Solomon |
| Biotechnology and Biological Sciences Research Council | R004765 | Samuel G Solomon Aman B Saleem |
| Human Frontier Science Program | RGY0076/2018 | Aman B Saleem |

The funders had no role in study design, data collection and interpretation, or the decision to submit the work for publication.

### Author contributions

Gonçalo Lopes, Conceptualization, Resources, Software, Formal analysis, Validation, Investigation, Visualization, Methodology, Writing - original draft, Writing - review and editing; Karolina Farrell, Validation, Writing - review and editing; Edward AB Horrocks, Mai M Morimoto, Amalia Papanikolaou, Fabio R Rodrigues, Thomas Wheatcroft, Stefano Zucca, Validation, Investigation, Writing - review and editing; Chi-Yu Lee, Validation; Tomaso Muzzu, Validation, Investigation, Writing - original draft, Writing - review and editing; Samuel G Solomon, Aman B Saleem, Conceptualization, Resources, Formal analysis, Supervision, Funding acquisition, Investigation, Visualization, Writing - original draft, Project administration, Writing - review and editing

## Author ORCIDs

Gonçalo Lopes  https://orcid.org/0000-0003-0731-4945
Karolina Farrell  https://orcid.org/0000-0002-0707-2838
Edward AB Horrocks  https://orcid.org/0000-0003-4019-5351
Chi-Yu Lee  https://orcid.org/0000-0001-6440-3050
Mai M Morimoto  https://orcid.org/0000-0002-9654-3960
Tomaso Muzzu  https://orcid.org/0000-0002-0018-8416
Amalia Papanikolaou  https://orcid.org/0000-0002-0048-6560
Fabio R Rodrigues  https://orcid.org/0000-0002-4848-7167
Thomas Wheatcroft  https://orcid.org/0000-0001-7990-3744
Stefano Zucca  https://orcid.org/0000-0002-8891-1225
Samuel G Solomon  https://orcid.org/0000-0001-5321-0288
Aman B Saleem  https://orcid.org/0000-0002-7100-1954

## Ethics

Human subjects: All procedures were approved by the Experimental Psychology Ethics Committee at University College London (Ethics Application EP/2019/002). We obtained informed consent, and consent to publish from all participants.

Animal experimentation: All experiments were performed in accordance with the Animals (Scientific Procedures) Act 1986 (United Kingdom) and Home Office (United Kingdom) approved project and personal licenses. The experiments were approved by the University College London Animal Welfare Ethical Review Board under Project License 70/8637.

## Decision letter and Author response

Decision letter https://doi.org/10.7554/eLife.65541.sa1
Author response https://doi.org/10.7554/eLife.65541.sa2

# Additional files

## Supplementary files

• Transparent reporting form

## Data availability

BonVision is an open-source software package available to use under the MIT license. It can be downloaded through the Bonsai (https://bonsai-rx.org) package manager, and the source code is available at: https://github.com/bonvision/BonVision. All benchmark programs and data are available at https://github.com/bonvision/benchmarks (copy archived at https://archive.softwareheritage.org/swh:1:rev:7205c04aa8fcba1075e9c9991ac117bd25e92639). Installation instructions, demos and learning tools are available at: https://bonvision.github.io/.

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

## Appendix 1

### Basic workflow structure

Each BonVision workflow starts by loading the basic Shaders library (this is Bonsai's implementation of OpenGL) and then creating a window in which stimuli are to be displayed. Bonsai is an event-based framework, so the visual stimulus generation and control are driven by events from the *RenderFrame* or *UpdateFrame* nodes, which are in turn activated when a screen refresh occurs. An event broadcast from the *RenderFrame* or *UpdateFrame* node then activates the cascade of nodes that load, generate, or update the different visual stimuli.

### Closed-loop control

Parameters of stimuli can also be updated, asynchronously and in parallel, by other events. Parameters of any Bonsai node can be controlled by addressing the relevant property within that node – all parameters within a node can be made visible to the external caller of that node. This is particularly useful for generating closed loop stimuli where the value of these parameters can be linked to external IO devices (e.g. position sensors) that are easily accessible using established Bonsai drivers and packages. A major advantage of the Bonsai framework is that the visual stimulus generation does not need to pause to poll those I/O devices, and the values from those devices can be retrieved any time up to the rendering of the frame, creating opportunities for low-lag updating of the visual stimulus.

### Considerations while using BonVision

#### Client control

Some experimental designs may rely on complex experimental control protocols that are already established in other software, or are challenging to implement in a reactive framework. For such applications, BonVision's rendering platform can be used as a client to create and control calibrated visual stimuli. This can be implemented using Bonsai's inbuilt IP communication protocols to interact with the independent controller software (e.g. Python or MATLAB). BonVision workflows can also be executed from the command-line using standard syntax, without opening the graphical interface of Bonsai.

#### Mercator projection

A key motivation in developing BonVision was the ability to present 2D and 3D stimuli in the same framework. To enable this, we chose to project 2D stimuli onto a 3D sphere, using the Mercator projection. The Mercator projection, however, contracts longitude coordinates around the two poles, and the consequence is that 2D stimuli presented close to the poles are deformed without compensation. Experiments that require 2D-defined stimuli to be presented near the default poles therefore need particular care. There are a few options to overcome this limitation. One option is to rotate the sphere mapping so that the poles are shifted away from the desired stimulus location. A second option is to present the texture on a 3D object facing the observer. For example, to present a grating in a circular aperture, we could have the grating texture rendered on a disk presented in 3D, and the disk is placed in the appropriate position. Finally, the user can present stimuli via the *NormalisedView* node, which defines stimuli in screen pixel coordinates, and use manual calibrations and precomputations to ensure the stimuli are of the correct dimensions.

#### Constructing 3D environments

There are many well-established software packages with graphical interfaces that are capable of creating 3D objects and scenes, and users are likely to have their preferred method. BonVision therefore focuses on providing easy importing of a wide variety of 3D model formats. BonVision offers three options for building 3D environments:

1. BonVision (limited capability). Inbuilt BonVision processes allow for the rendering of textures onto simple planar surfaces. The user defines the position and orientation of each plane in 3D

space, and the texture that is to be drawn onto that plane, using the *DrawTexturedModel* node.

2. Import (load) 3D models of objects (including cubes, spheres, and more complex models). Common 3D models (such as those used in *Figure 1*) are often freely available online. Custom models can be generated using standard 3D software, including Blender and CAD programmes. The user defines the position of each object, and its dynamics, within BonVision, and can independently attach the desired texture(s) to each of the different faces of those objects using the *DrawTexturedModel* node.

3. Import a full 3D scene (with multiple objects and camera views). BonVision is able to interact with both individual objects and cameras defined within a 3D scene. A particular advantage of this method is that specialised software (e.g. Blender) provide convenient methods to construct and visualise scenes in advance; BonVision provides the calibrated display environment and capacity for interaction with the objects.

Once the 3D scene is created, the user can then control a camera (e.g. move or rotate) in the resultant virtual world. BonVision computes the effects of the camera movement (i.e. without any additional user code) to render what the camera should see onto a display device.

## Animation lags and timing logs

While BonVision expends substantial effort to eliminate interruptions to the presentation of a visual stimulus, these can occur, and solutions may be beyond the control of the experimenter. To avoid the potential accumulation of timing errors, the UpdateFrame node uses the current time to specify the current location in an animation sequence. The actual presentation time of each frame in an animation can be logged using the standard logging protocols in BonVision. The log can also include the user predefined or real-time updated parameters that were used to generate the corresponding stimulus frame.

## Customised nodes and new stimuli

Bonsai's modular nature and simple integration with C# and Python scripting means BonVision can be extended by users. The BonVision package is almost entirely implemented using the Bonsai visual programming language, showcasing its power as a domain-specific language. Custom BonVision nodes can be easily created in the graphical framework, or using C# or Python scripting with user-defined inputs, outputs, properties and operations can be generated by users to create novel visual stimuli, define interactions between objects and enable visual environments which are arbitrarily responsive to experimental subjects.

## Physics engine

BonVision is able to calculate interactions between objects using the package Bonsai.Physics, including collisions, bouncing off surfaces, or deformations.

## Spatial calibration

BonVision provides automatic calibration protocols to define the position of display(s) relative to the observer. A single positional marker is sufficient for each flat display (illustrated in *Figure 2*; a standard operating procedure is described on the website). An additional marker is placed in the position of the observer to provide the reference point.

When the observer's position relative to the display varies (e.g. in the augmented reality example in *Figure 3* and *Figure 3—video 1*), the easiest solution is to calibrate the position of the displays relative to a fixed point in the arena. The observer position is then calculated in real-time, and the vector from the observer to the reference point is added to the vector from the reference to the display. The resultant vector is the calibrated position of the display relative to the observer's current position.

In the case of head-mounted displays (HMDs), BonVision takes advantage of the fact that HMD drivers can provide the calibrated transform matrices from the observer's eye centre, using the *HMDView* node.

When the presentation surface is curved (e.g. projection onto a dome) a manual calibration step is required as in other frameworks. This calibration step is often referred to as mesh-mapping and involves the calculation of a transformation matrix that specifies the relationship between a (virtual) flat display and position on the projection surface. A standard operating procedure for calculating this mesh-map is described on the BonVision website.

## Performance optimisation

We recommend displaying stimuli through a single graphics card when possible. When multiple displays are used for visual stimulation, we recommend configuring them as a single extended display (as seen by the operating system). All our tests were performed under this configuration.

**Appendix 1—table 1.** Features of visual display software.

| Features | BonVision | PsychToolbox | PsychoPy | ViRMEn | ratCAVE | FreemoVR | Unity |
|---|---|---|---|---|---|---|---|
| Free and Open-source (FOSS) | √√ | √# | √√ | √# | √ | √√ | √ |
| Rendering of 3D environments | √√ | √ | √ | √√ | √√ | √√ | √√ |
| Dynamic rendering based on observer viewpoint | √√ | | | √ | √√ | √√ | √ |
| GUI for designing 3D scenes | | | | √√ | | | √√ |
| Import 3rd party 3D scenes | √√ | √ | √ | | | | √√ |
| Real-time interactive 3D scenes | √√ | √ | | √√ | √√ | √√ | √√ |
| Web-based deployment | | | √√ | | | | √√ |
| Interfacing with cameras, sensors and effectors | √√ | √√ | ~ | √√ | | ~ | ~ |
| Real-time hardware control | √√ | ~ | ~ | √ | √√ | √ | √ |
| Traditional visual stimuli | √√ | √√ | √√ | | | | |
| Auto-calibration of display position and pose | √√ | | | | | | |
| Integration with deep learning pose estimation | √√ | | | | | | |

√√ easy and well-supported.

√ possible, not well-supported.

~ difficult to implement.

# based on MATLAB (requires a license).

## Learning to use BonVision

We provide the following learning materials (which will continue to be updated):

> Tutorials and Documentation: https://bonvision.github.io
> Video tutorials: https://www.youtube.com/channel/UCEg-3mfbvjIwbzDVvqYudAA
> Demos and Examples: https://github.com/bonvision/examples
> Community forum: https://groups.google.com/forum/#!forum/bonsai-users

