## [Decision Letter]

**Acceptance summary:**

Increasingly, neuroscience experiments require immersive virtual environments that approximate natural sensory motor loops while permitting high-bandwidth measurements of brain activity. BonVision is an open-source graphics programming library that allows experimenters to quickly implement immersive 3D visual environments across display hardware and geometry with automated calibration and integration with hundreds of different neural recording technologies, behavioral apparatuses, etc. BonVision standardizes sharing complex, closed-loop visual tasks between labs with vastly different equipment and provides a concrete and easy way to do so.

**Decision letter after peer review:**

Thank you for submitting your article "Creating and controlling visual environments using BonVision" for consideration by *eLife*. Your article has been reviewed by 3 peer reviewers, and the evaluation has been overseen by Chris Baker as the Senior and Reviewing Editor. The following individuals involved in review of your submission have agreed to reveal their identity: Jonathan P Newman (Reviewer #1); André Maia Chagas (Reviewer #2); Sue Ann Koay (Reviewer #3).

The reviewers have discussed their reviews with one another, and the Reviewing Editor has drafted this letter to help you prepare a revised submission.

Essential revisions:

In general, the reviewers were very positive about the manuscript and appreciated the time and effort taken both to develop BonVision and write this manuscript. The major concerns reflect a desire from the reviewers to see more detail on specific points as well as clarification over some of the statements made.

In your revision please address the specific recommendations below.

*Reviewer #1 (Recommendations for the authors):*

General comment: There are two measures of performance that are not explored in the manuscript but may aid in describing BonVision's advantages over alternative software. The first is the improved performance and ease of use compared to alternatives in cases where the input used to drive visual stimuli consists of mixtures of asynchronous data sources (e.g. ephys and behavioral measurements together). This is something I imagine BonVision could do with less effort and greater computational efficiency than alternative software. The animal experiments provided are good benchmarks because they are common designs, but do not demonstrate BonVision's immense potential for easily creating visual tasks with complex IO and stimulus contingencies. The second is a measure of human effort required to implement an experiment using BonVision compared to imperative, text-based alternatives. I think both of these issues could be tackled in the discussion by expanding a bit on Lines 144-146: why are BonVision and Bonsai so good at data stream composition compared to alternatives, and why is a visual programming approach so appropriate for Bonsai/BonVision's target use cases?

General comment: Following up on my desire for a more detailed explanation of the operation of the Bonsai.Shaders library, most of its operators have obvious relations to traditional OpenGL programming operations. However, an explanation of how the traditionally global state machine (context) of OpenGL was mapped onto the Bonsai.Shaders nodes and how temporal order of OpenGL context option manipulation is enforced might be helpful for those wishing to understand the underlying mechanics of BonVision and create their own tools using the Shaders library.

Line 11: The use of the word "timing" is ambiguous to me. Are the authors referring to closed loop reaction times and determinism, hardware IO delays, the combination of samples from asynchronous data streams, or all of the above?

Lines 13 and 22: The authors correctly state that graphics programming requires advanced training. However, the use of Bonsai, a functional language that operates entirely on Observable Sequences, also requires quite a lot of training to use effectively. I do think the authors have a point here, and I agree Bonsai is tool worth learning, but I feel the main strength of using Bonsai is its (broadly defined) performance (speed, elegance when dealing with asynchronous data, ease and formality of experiment sharing, ease of rapid prototyping, etc) rather than its learning curve. This point is exacerbated by the lack of documentation (outside of the source code) for many Bonsai features.

Line 64: Adding a parenthetical link to the Blender website seems appropriate.

Line 97: The model species should be stated here.

Figure 4(C): There is a single instance of BonVision being outperformed by PsychoPy3 in the case of 16-layer texture blending at 60 FPS. Can the authors comment on why this might be (e.g. PsychoPy3's poor performance at low layer counts is due to some memory bottleneck?) and why this matters (or does not matter) practically in the context of BonVision's target uses?

Figure 4(A-C): The cartoons of display screens have little black boxes in the lower right corners and I'm not sure what they mean.

Figure 5(A): As mentioned previously, it seems that these are post-hoc temporally aligned receptive fields (RFs). Is it worth seeing what the RFs created without post-hoc photodiode-based alignment of stimuli onset look like so that we can see the effect of display presentation jitter (or lack thereof)? This would be a nice indication of the utility of the package for real-time stimulus shaping for system ID purposes where ground truth alignment might not be possible. This is made more relevant given BonVision's apparent larger latency jitter compared to PsychToolbox (Figure 4A).

Figure 5(D): Although useful, the size discrimination task probably does not cover all potential corner cases with this type of projection. I don't think more experiments need to be performed but a more thorough theoretical comparison with other methods, e.g. sphere mapping, might be useful to motivate the choice of cube mapping for rendering 3D objects, perhaps in the discussion.

Figure 5(I): The caption refers to the speed of the animal on the ordinate axis but that figure seems to display a psychometric curve for freezing or running behaviors over time from stimulus presentation.

Line 293 and 319-322: HARP is a small enough project that I feel some explanation of the project's intentions and capabilities and the Bonsai library used to acquire it from HARP hardware might be useful.

Line 384: "OpenEphys" should be changed to "Open Ephys" in the text and in reference 13 used to cite the Acquisition Board's use.

*Reviewer #2 (Recommendations for the authors):*

– Figures 1, 2, 3 and Supp1 – the indication of the observer in these figures is sometimes a black, and sometimes a red dot. This is not bad, but I think you could streamline your figures if on the first one you had a legend for what represents the observer (ie observer = red dot) and have the same pattern through the figures?

– Figure 2 – If I understand correctly, in panels C and E, the markers are read by an external camera, which I am supposing in this case are the laptop camera? If this is the case, could you please change these panels so that they explicitly show where the cameras are? Maybe adding the first top left panel from supp Figure 3 to Figure 2 and indicate from where the markers are read would solve this?

– Figure 5 – Panel I: the legend states "The speed of the animal across different trials, aligned to the time of stimulus appearance." but the figure Y axis states Cumulative probability. I guess the legend needs updating? Also it is not clear to me how the cumulative probabilities of freeze and flight can sum up to more than one, as it seems to be the case from the figure? I am assuming that an animal either freezes of flees in this test? Maybe I missed something?

– In the results, lines 40 to 42, the authors describe how they have managed to have a single framework for both traditional visual presentation and immersive virtual reality. Namely they project 2D coordinate frame to a 3D sphere using Mercator projection. I would like to ask the authors to explain a bit how they deal with the distortions present in this type of projection. As far as I understand, this type of projection inflates the size of objects that are further away from the sphere midline (with increased intensity the further away)? Is this compensated for in the framework somehow? Would it make sense to offer users the option to choose different projections depending on their application?

– In line 62 "BonVision also has the ability to import standard format 3D design files" could the authors specify which file formats are accepted?

– When benchmarking BonVision (starting on line 73), the authors focus on 60Hz stimulus presentation using monitors with different capabilities. This is great, as it addresses main points for human, non-human primates and rodent experiments. I believe however that it would be great for the paper and the community in general if the authors could do some benchmarking with higher frame rates and contextualize BonVision for the use with other animal models, such as Fly, fish, etc. Given that there are a couple of papers describing visual stimulators that take care of the different wavelengths needed to stimulate the visual system of these animals, it seems to me that BonVision would be a great tool to create stimuli and environments for these stimulators and animal models.

*Reviewer #3 (Recommendations for the authors):*

I have a few presentation style points where I feel the text should be more careful not to come across as unintendedly too strong, or otherwise justification need to be provided to substantiate the claims. Most importantly, line 19 "the ability for rapid and efficient interfacing with external hardware (needed for experimentation) without development of complex multi-threaded routines" is a bit mysterious to me because I am unsure what these external hardware are that BonVision facilitates interfacing with. For example, experimenters do prefer multi-threaded routines where the other threads are used to trigger reward delivery, sensory stimuli of other modalities, or control neural stimulation or recording devices. This is in order to avoid blocking execution of the visual display software when these other functions are called. If BonVision provides a solution for these kinds of experiment interfacing requirements, I think they are definitely important enough to mention in the text. Otherwise, the sentence of line 19 needs some work in order to make it clear as to exactly which functionalities of BonVision are being referred to.

The other claims that stood out to me are as follows. In the abstract it is said that "Real-time rendering… necessary for next-generation…", but I don't know if anybody can actually claim that any one method is necessary. In line 116, "suggesting habituation to the virtual environment", the authors can also acknowledge that mice might simply be habituating to the rig (e.g. even if there was no visual display), since this does not seem to be a major claim that needs to be made. The virtual cliff effect (line 118) also seems very interesting, but the authors have not fully demonstrated that mice are not alternatively responding to a change in floor texture. It is also unclear to me why a gray floor (which looks to be equiluminant with the rest of the textured floor at least by guessing from Figure 5F) should be visually identified as a cliff, as opposed to, say, black. In order to make this claim about visual cliff identification especially without binocular vision, the authors would probably have to show experiments where the mice do not slow down at other floor changes (to white maybe?), but I'm unsure as to whether the data exists for this nor whether it is worth the effort. Overall I don't see a reason why the authors should attempt to claim that "BonVision is capable of eliciting naturalistic behaviors in a virtual environment", since the naturalness of rodent behaviors in virtual environments is a topic of debate in some circles, independent of the software used to generate those environments. I figure it's better to stay away unless this is a fight that one desires to fight.

---

## [Author Response]

Reviewer #1 (Recommendations for the authors):General comment: There are two measures of performance that are not explored in the manuscript but may aid in describing BonVision's advantages over alternative software. The first is the improved performance and ease of use compared to alternatives in cases where the input used to drive visual stimuli consists of mixtures of asynchronous data sources (e.g. ephys and behavioral measurements together). This is something I imagine BonVision could do with less effort and greater computational efficiency than alternative software. The animal experiments provided are good benchmarks because they are common designs, but do not demonstrate BonVision's immense potential for easily creating visual tasks with complex IO and stimulus contingencies. The second is a measure of human effort required to implement an experiment using BonVision compared to imperative, text-based alternatives. I think both of these issues could be tackled in the discussion by expanding a bit on Lines 144-146: why are BonVision and Bonsai so good at data stream composition compared to alternatives, and why is a visual programming approach so appropriate for Bonsai/BonVision's target use cases?

We agree and we have now revised the Introduction and Discussion to better make these points transparent (particularly around lines 44-55 and 235-239).

General comment: Following up on my desire for a more detailed explanation of the operation of the Bonsai.Shaders library, most of its operators have obvious relations to traditional OpenGL programming operations. However, an explanation of how the traditionally global state machine (context) of OpenGL was mapped onto the Bonsai.Shaders nodes and how temporal order of OpenGL context option manipulation is enforced might be helpful for those wishing to understand the underlying mechanics of BonVision and create their own tools using the Shaders library.

We thank the reviewer for prompting us. Generally, we now mention that we build on the Bonsai.Shaders package in new text (lines 58-59 and in Supplementary Details).

Specifically, related to the point related to the temporal order of OpenGL, we also include the text (in lines 133-135): “BonVision accumulates a list of the commands to OpenGL as the program makes them. To optimise rendering performance, the priority of these commands is ordered according to that defined in the Shaders component of the *LoadResources* node (which the user can manipulate for high-performance environments). These ordered calls are then executed when the frame is rendered.”

Line 11: The use of the word "timing" is ambiguous to me. Are the authors referring to closed loop reaction times and determinism, hardware IO delays, the combination of samples from asynchronous data streams, or all of the above?

Thank you for picking this up – the organisation of the first paragraph meant that the subject of this sentence was unclear, and we have now tried to make this paragraph clearer, including splitting it into two distinct points (lines 3-17). We hope these changes now address the reviewers point.

Lines 13 and 22: The authors correctly state that graphics programming requires advanced training. However, the use of Bonsai, a functional language that operates entirely on Observable Sequences, also requires quite a lot of training to use effectively. I do think the authors have a point here, and I agree Bonsai is tool worth learning, but I feel the main strength of using Bonsai is its (broadly defined) performance (speed, elegance when dealing with asynchronous data, ease and formality of experiment sharing, ease of rapid prototyping, etc) rather than its learning curve. This point is exacerbated by the lack of documentation (outside of the source code) for many Bonsai features.

We agree and have revised the Introduction (lines 42-55) and Discussion (lines 235-239) to make this clearer.

Line 64: Adding a parenthetical link to the Blender website seems appropriate.Line 97: The model species should be stated here.

Done.

Figure 4(C): There is a single instance of BonVision being outperformed by PsychoPy3 in the case of 16-layer texture blending at 60 FPS. Can the authors comment on why this might be (e.g. PsychoPy3's poor performance at low layer counts is due to some memory bottleneck?) and why this matters (or does not matter) practically in the context of BonVision's target uses?

In the conditions under which the benchmarking was performed, PsychoPy was able to present more overlapping stimuli compared to BonVision and PsychToolBox, because PsychoPy presented stimuli at a lower resolution compared to the other systems. We now indicate this in the main text of the manuscript (lines 150-151).

Figure 4(A-C): The cartoons of display screens have little black boxes in the lower right corners and I'm not sure what they mean.

The black square represents the position of a flickering square, the luminance of which is detected by a photodiode and used to measure frame display times. We have now updated the legend of Figure 4 to make this clear.

Figure 5(A): As mentioned previously, it seems that these are post-hoc temporally aligned receptive fields (RFs). Is it worth seeing what the RFs created without post-hoc photodiode-based alignment of stimuli onset look like so that we can see the effect of display presentation jitter (or lack thereof)? This would be a nice indication of the utility of the package for real-time stimulus shaping for system ID purposes where ground truth alignment might not be possible. This is made more relevant given BonVision's apparent larger latency jitter compared to PsychToolbox (Figure 4A).

We thank the reviewer for this suggestion. We now include receptive field maps calculated using the BonVision timing log in Figure5—figure supplement 1. Using the BonVision timing alone was also effective in identifying receptive fields.

Figure 5(D): Although useful, the size discrimination task probably does not cover all potential corner cases with this type of projection. I don't think more experiments need to be performed but a more thorough theoretical comparison with other methods, e.g. sphere mapping, might be useful to motivate the choice of cube mapping for rendering 3D objects, perhaps in the discussion.

We now clarify that we use the size discrimination task as a simple test of the ability of BonVision to run VR stimuli on a head-mounted display.

Although we considered the different mapping styles, we settled on cube mapping for 3D stimuli, as this is currently the standard for 3D rendering systems, and the most computationally efficient. We included a detailed discussion on the merits and issues with Mercatore projection for 2D stimuli in the new section “Appendix 1”.

Figure 5(I): The caption refers to the speed of the animal on the ordinate axis but that figure seems to display a psychometric curve for freezing or running behaviors over time from stimulus presentation.

Thank you for pointing this out, we have now corrected this.

Line 293 and 319-322: HARP is a small enough project that I feel some explanation of the project's intentions and capabilities and the Bonsai library used to acquire it from HARP hardware might be useful.

We have now added more information on the HARP sources, and why we have employed it here, including details of the Bonsai library needed to use the HARP device (lines 648-652). However, we are not core members of the HARP project and are wary of speaking for them on its intentions and other capabilities.

Line 384: "OpenEphys" should be changed to "Open Ephys" in the text and in reference 13 used to cite the Acquisition Board's use.

Done.

Reviewer #2 (Recommendations for the authors):– Figures 1, 2, 3 and Supp1 – the indication of the observer in these figures is sometimes a black, and sometimes a red dot. This is not bad, but I think you could streamline your figures if on the first one you had a legend for what represents the observer (ie observer = red dot) and have the same pattern through the figures?

Great suggestion, thank you. We have now changed all observers to red dots and indicated this in the legend.

– Figure 2 – If I understand correctly, in panels C and E, the markers are read by an external camera, which I am supposing in this case are the laptop camera? If this is the case, could you please change these panels so that they explicitly show where the cameras are? Maybe adding the first top left panel from supp Figure 3 to Figure 2 and indicate from where the markers are read would solve this?

We think that the reviewer had spotted that there are multiple cameras shown in the image, and we apologise for not spotting this ourselves. The calibration is performed using only the images shown (that is the camera that is taking the image is the one used for the calibration). We now make this clearer in the legend to Figure 2.

– Figure 5 – Panel I: the legend states "The speed of the animal across different trials, aligned to the time of stimulus appearance." but the figure Y axis states Cumulative probability. I guess the legend needs updating? Also it is not clear to me how the cumulative probabilities of freeze and flight can sum up to more than one, as it seems to be the case from the figure? I am assuming that an animal either freezes of flees in this test? Maybe I missed something?

We thank the reviewer for highlighting this error. We have updated the legend.

– In the results, lines 40 to 42, the authors describe how they have managed to have a single framework for both traditional visual presentation and immersive virtual reality. Namely they project 2D coordinate frame to a 3D sphere using Mercator projection. I would like to ask the authors to explain a bit how they deal with the distortions present in this type of projection. As far as I understand, this type of projection inflates the size of objects that are further away from the sphere midline (with increased intensity the further away)? Is this compensated for in the framework somehow? Would it make sense to offer users the option to choose different projections depending on their application?

This is an excellent point. We have added a specific discussion related to the Mercator projection in the new section called Appendix, where we discuss the distortions and methods to work around them.

– In line 62 "BonVision also has the ability to import standard format 3D design files" could the authors specify which file formats are accepted?

We now link from the main text to the ‘File Formats’ section in Methods.

– When benchmarking BonVision (starting on line 73), the authors focus on 60Hz stimulus presentation using monitors with different capabilities. This is great, as it addresses main points for human, non-human primates and rodent experiments. I believe however that it would be great for the paper and the community in general if the authors could do some benchmarking with higher frame rates and contextualize BonVision for the use with other animal models, such as Fly, fish, etc. Given that there are a couple of papers describing visual stimulators that take care of the different wavelengths needed to stimulate the visual system of these animals, it seems to me that BonVision would be a great tool to create stimuli and environments for these stimulators and animal models.

We have added a new Figure 4—figure supplement 1, in which we show the results of the non-overlapping textures benchmark for BonVision at 144 Hz refresh. Comparison with the same data obtained at 60 Hz shows little deterioration in performance. These new data supplement the extant tests in Figure 4A, where we tested the closed-loop latency at these higher frame rates.

Reviewer #3 (Recommendations for the authors):I have a few presentation style points where I feel the text should be more careful not to come across as unintendedly too strong, or otherwise justification need to be provided to substantiate the claims. Most importantly, line 19 "the ability for rapid and efficient interfacing with external hardware (needed for experimentation) without development of complex multi-threaded routines" is a bit mysterious to me because I am unsure what these external hardware are that BonVision facilitates interfacing with. For example, experimenters do prefer multi-threaded routines where the other threads are used to trigger reward delivery, sensory stimuli of other modalities, or control neural stimulation or recording devices. This is in order to avoid blocking execution of the visual display software when these other functions are called. If BonVision provides a solution for these kinds of experiment interfacing requirements, I think they are definitely important enough to mention in the text. Otherwise, the sentence of line 19 needs some work in order to make it clear as to exactly which functionalities of BonVision are being referred to.

We agree and have now revised the Introduction (lines 42-55) to make these points clearer.

The other claims that stood out to me are as follows. In the abstract it is said that "Real-time rendering… necessary for next-generation…", but I don't know if anybody can actually claim that any one method is necessary.

We have changed the text to say ‘important’ rather than ‘necessary’.

In line 116, "suggesting habituation to the virtual environment", the authors can also acknowledge that mice might simply be habituating to the rig (e.g. even if there was no visual display), since this does not seem to be a major claim that needs to be made. The virtual cliff effect (line 118) also seems very interesting, but the authors have not fully demonstrated that mice are not alternatively responding to a change in floor texture. It is also unclear to me why a gray floor (which looks to be equiluminant with the rest of the textured floor at least by guessing from Figure 5F) should be visually identified as a cliff, as opposed to, say, black. In order to make this claim about visual cliff identification especially without binocular vision, the authors would probably have to show experiments where the mice do not slow down at other floor changes (to white maybe?), but I'm unsure as to whether the data exists for this nor whether it is worth the effort. Overall I don't see a reason why the authors should attempt to claim that "BonVision is capable of eliciting naturalistic behaviors in a virtual environment", since the naturalness of rodent behaviors in virtual environments is a topic of debate in some circles, independent of the software used to generate those environments. I figure it's better to stay away unless this is a fight that one desires to fight.

We agree that there are heated debates around these issues in the field, and that this is not the place to have those discussions. We have changed the relevant sentence to read (lines 204-205): “BonVision is therefore capable of generating virtual reality environments which both elicit, and are responsive to animal behaviour.”